# Application of Two-Dimensional Fluorescence Spectroscopy for the On-Line Monitoring of Teff-Based Substrate Fermentation Inoculated with Certain Probiotic Bacteria

**DOI:** 10.3390/foods11081171

**Published:** 2022-04-18

**Authors:** Sendeku Takele Alemneh, Shimelis Admassu Emire, Mario Jekle, Olivier Paquet-Durand, Almut von Wrochem, Bernd Hitzmann

**Affiliations:** 1Department of Process Analytics and Cereal Science, Institute of Food Science and Biotechnology, University of Hohenheim, 70599 Stuttgart, Germany; sendeku.tekele@aait.edu.et (S.T.A.); o.paquet-durand@uni-hohenheim.de (O.P.-D.); almut.vonwrochem@uni-hohenheim.de (A.v.W.); 2Food Engineering, Addis Ababa Institute of Technology, Addis Ababa University, Addis Ababa 1000, Ethiopia; shimelis.admassu@aait.edu.et; 3Department of Plant-Based Foods, Institute of Food Science and Biotechnology, University of Hohenheim, 70599 Stuttgart, Germany; mario.jekle@uni-hohenheim.de

**Keywords:** artificial neural network, functional beverage, partial least-squares regression, probiotics, teff-based substrate, 2D-fluorescence spectroscopy

## Abstract

There is increasing demand for cereal-based probiotic fermented beverages as an alternative to dairy-based products due to their limitations. However, analyzing and monitoring the fermentation process is usually time consuming, costly, and labor intensive. This research therefore aims to apply two-dimensional (2D)-fluorescence spectroscopy coupled with partial least-squares regression (PLSR) and artificial neural networks (ANN) for the on-line quantitative analysis of cell growth and concentrations of lactic acid and glucose during the fermentation of a teff-based substrate. This substrate was inoculated with mixed strains of *Lactiplantibacillus plantarum* A6 (LPA6) and *Lacticaseibacillus rhamnosus* GG (LCGG). The fermentation was performed under two different conditions: condition 1 (7 g/100 mL substrate inoculated with 6 log cfu/mL) and condition 2 (4 g/100 mL substrate inoculated with 6 log cfu/mL). For the prediction of LPA6 and LCGG cell growth, the relative root mean square error of prediction (pRMSEP) was measured between 2.5 and 4.5%. The highest pRMSEP (4.5%) was observed for the prediction of LPA6 cell growth under condition 2 using ANN, but the lowest pRMSEP (2.5%) was observed for the prediction of LCGG cell growth under condition 1 with ANN. A slightly more accurate prediction was found with ANN under condition 1. However, under condition 2, a superior prediction was observed with PLSR as compared to ANN. Moreover, for the prediction of lactic acid concentration, the observed values of pRMSEP were 7.6 and 7.7% using PLSR and ANN, respectively. The highest error rates of 13 and 14% were observed for the prediction of glucose concentration using PLSR and ANN, respectively. Most of the predicted values had a coefficient of determination (R^2^) of more than 0.85. In conclusion, a 2D-fluorescence spectroscopy combined with PLSR and ANN can be used to accurately monitor LPA6 and LCGG cell counts and lactic acid concentration in the fermentation process of a teff-based substrate. The prediction of glucose concentration, however, showed a rather high error rate.

## 1. Introduction

Consumer demand for probiotic fermented cereal-based beverages is increasing. This is predominantly due to the limitations associated with dairy-based products, i.e., lactose and milk protein sensitivity or intolerance, fat content, and consumers’ desire for foods without animal products [1]. However, analyzing and monitoring the fermentation process with conventional methods such as high-performance liquid chromatography is challenging, as it is time-consuming, costly, and labor-intensive [2]. Further challenges may arise during sterilization, calibration, and sampling [3]. The productivity of the fermentation process and the cost of the fermented products depend mostly on the methods of monitoring and on the control over the operating conditions. On-line control of the fermentation process usually involves determination of pH and temperature. However, other key parameters of the fermentation process such as concentrations of metabolite, substrate, and cellular density need further investigation [4]. The key fermentation process parameters are not usually examined due to the expensive and time-consuming measurement methods [5].

Glucose was found to be the primarily consumed substrate while lactic acid was the primary metabolic product observed in the fermentation process of teff-based substrates inoculated with *Lactiplantibacillus plantarum A6* (LPA6) and *Lacticaseibacillus rhamnosus GG* (LCGG) [6]. Teff is a staple food crop in Ethiopia and Eritrea. It is gluten-free and an attractive source of iron (363 mg/kg flour) [7]. Thus, the fermentation industry needs an effective and efficient method to supervise the fermentation process. On-line analysis of the key fermentation process parameters assures product quality and productivity [8].

An alternative approach for this purpose is the application of a 2D-fluorescence spectroscopy. It is an ideal instrument for the on-line supervision of fermentation processes. In addition, its measurement is non-invasive and does not interfere with the fermentation medium [9]. In a 2D-fluorescence measurement, many wavelength combinations of excitation and emission are measured. A large volume of spectral data can be evaluated quantitatively using chemometric methods such as principal component regression (PCR) and partial least-squares regression (PLSR) [10]. Calibrating multivariate spectral data for quantitative spectral evaluation is becoming a standard method which allows an examination of several analytes at the same time. The PLSR and PCR are full-spectrum and are factor analyses based on multivariate calibration methods [11]. While PLSR and PCR are the most widely employed chemometric methods, PLSR usually requires fewer latent variables than PCR without influencing its predictive ability. Moreover, PLSR has superior prediction ability to PCR when there are different independent spectral components that can join with the spectral features [12].

Another method for the evaluation of the spectral data involves the use of artificial neural networks (ANN), which can be used to model a nonlinear correlation of the spectra with the variables [13]. ANN models generally contain two or more layers, each having a number of neurons. The ANN’s activation functions are used to connect the neurons of the different layers to each other. One vital process in the utilization of ANN is training. It serves to minimize errors between the model output and measured values. The process of training is a continual one and consists of adjusting biases and weights at each sequence. The training process is completed when the error rate is at its lowest [14].

As of yet, there has been no work conducted on the simultaneous measurements of cell growth, glucose, and lactic acid in samples from a fermented teff-based substrate inoculated with mixed strains of LPA6 and LCGG using 2D-fluorescence spectroscopy. Therefore, this research aims to apply the potential of 2D-fluorescence spectroscopy and mathematical models of PLSR and ANN as tools for the on-line analysis of the fermentation process of a teff-based substrate inoculated with co-culture strains of LPA6 and LCGG.

## 2. Materials and Methods

### 2.1. Materials

Whole-grain teff flour was purchased from Teff-shop.de, Manuel Boesel, Homburger Str.49a, 61191 Rosbach von der Höhe, Germany. Freeze-dried strains of LPA6 (LMG 18053, BCCM, Gent, Belgium) and LCGG (LMG 18243, BCCM, Gent, Belgium) were provided by the Department of Process Analytics and Cereal Science, Institute of Food Science and Biotechnology, University of Hohenheim, Stuttgart, Baden-Württemberg, Germany.

### 2.2. Starter Culture Preparation and Storage

Freeze-dried strains of LPA6 (LMG 18053, BCCM, Gent, Belgium) and LCGG (LMG 18243, BCCM, Gent, Belgium) were activated and placed in a refrigerator (6 °C) until the inoculation of the fermentation medium. The starter culture of LPA6 was prepared using the method described by Bationo, Songré-Ouattara, Hemery, Hama-Ba, Parkouda, Chapron, Le Merrer, Leconte, Sawadogo-Lingani, and Diawara [15]. The inoculum of LPA6 was obtained with sterilized MRS (DE MAN, ROGOSA, and SHARPE) broth by incubating in an incubator (BINDER GmbH, KB 115, Tuttlingen, Germany) for 24 h at 30 °C. Furthermore, the LCGG starter culture was prepared according to the method used by Matejčeková, Liptáková, and Valík [16]. The LCGG inoculum was collected after 24 h of incubation at 37 °C in sterilized MRS broth. Starter cultures were harvested by centrifugation (Mega star 600R, Leuven, Belgium) at 3000× *g*, 4 °C for 15 min. The LPA6 and LCGG cells were washed using a sterilized saline solution (0.9% NaCl). Finally, the supernatant was removed, and cell pellets were mixed with a sterilized saline solution to form a cell suspension of approximately 9 log cfu/mL. This was taken as an inoculum and was kept in a refrigerator (6 °C) until utilization within 48 h.

Strains of LPA6 and LCGG were stored with a medium containing 60% MRS broth and 40% glycerol in a deep freezer (−70 °C) [17].

### 2.3. Off-Line Measurement of Microbial Viability

The LPA6 cell count was determined by counting individual colonies on MRS agar plates (colony-forming units—cfu) according to the method used by Alemneh, Emire, and Hitzmann [6]. Each reported value represents the mean count of three plates containing 25–250 colonies. Plate agar was made by mixing 15 g agar into 1 L of MRS broth. Serial ten-fold dilutions of samples were prepared using a 0.9% NaCl solution. Fifty μL drops of diluted samples were put on MRS agar plates and incubated overnight at 30 °C. A similar procedure was followed for counting LCGG cells. However, the incubation time was nearly 48 h. For counting LPA6 and LCGG cells, the method developed by Alemneh, Emire, and Hitzmann [6] was used. Samples with co-culture strains of LPA6 and LCGG were incubated for 48 h on MRS agar plates at 30 °C. After overnight incubation, LPA6 cells were counted. Afterwards, LCGG was grown for approximately 48 h of incubation. Then, total cell counts of both LPA6 and LCGG were recorded. Finally, the difference (total cell counts of LPA6 and LCGG—cell count of LPA6) was determined as the LCGG cell count.

### 2.4. Fermentation Process Conditions

Overnight cultures of LPA6 and LCGG, each with an initial cell density of 6 log cfu/mL, were inoculated to the fermenting substrates, which were prepared from 4 and 7 g/100 mL of whole-grain teff flour in distilled water. Two different fermentation conditions were examined: condition 1, 7 g/100 mL substrate inoculated with 6 log cfu/mL mixed strains of LPA6 and LCGG and condition 2, 4 g/100 mL substrate inoculated with 6 log cfu/mL mixed strains of LPA6 and LCGG. Before fermentation, the substrates were heated in a water bath (GFL-1083, Burgwedel, Germany) set at 85 °C for 15 min and then sterilized in an autoclave (SHP Laboklav, 160-MSLV, Satuelle, Germany). Before the addition of microbes, the sterilized substrates were cooled down in a safety cabinet (Kendro Laboratory Products GmbH, KS 12, Hanau, Germany). Fermentations were performed without pH control for 15 h using a 2.5 L bioreactor (INFORS AG CH-4103, Bottmingen, Switzerland). The working volume was 1 L, the stirrer speed of the bioreactor was 150 rpm, and the fermentation temperature was 37 °C. Three h after the start of fermentation, samples were taken at 2 h intervals for determining LPA6 and LCGG cell counts and analyzing the concentration of glucose and lactic acid.

### 2.5. Off-Line Measurement of Glucose and Lactic Acid

Glucose and lactic acid content was determined using high-performance liquid chromatography (HPLC). Samples were centrifuged at 3000× *g*, 4 °C for 15 min and the supernatant filtered with a 0.45 μm polypropylene membrane (VWR, Darmstadt, Germany). After filtration, samples were analyzed by HPLC (ProStar, Variant, Walnut Creek, CA, USA), which was equipped with a refractive index detector. Twenty μL of samples were injected into a Rezex ROA-organic acid H^+^ (8%) column (Phenomenex, Aschaffenburg, Germany). The working temperature was set at 70 °C, and a 5 mM H_2_SO_4_ solvent with a flow rate of 0.6 mL/min was used. Lactic acid and glucose content was obtained using Software Galaxie^TM^ Chromatography (Varian, Walnut Creek, CA, USA). Duplicate measurements were obtained for each analyte.

### 2.6. On-Line Measurement Using 2D-Fluorescence Spectroscopy

A BioView sensor (DELTA Lights and Optics, Venlighedsvej 4, 2970, Horsholm, Denmark) was used to collect 2D-fluorescence spectra. A fluorescence probe was attached to the sterilized bioreactor over a light guide, which connected with a 25 mm standard port. This standard port has a quartz glass window to interface with the bioreactor. Therefore, there was no contact between the fermenting medium and the actual probe. A BioView sensor measured several combinations of excitation (270–550) and emission (310–590). The observed fluorescence spectrum had 120 intensity values of wavelength combinations measured in intervals of 20 nm. Off-line measured results and the analogous fluorescence spectra data were utilized to develop calibration models of PLSR and ANN for the prediction of LPA6 and LCGG cell counts and concentrations of glucose and lactic acid. Software Unscrambler X version 10.3 (CAMO Software AS., Oslo, Norway) and MATLAB R2019a version 9.6 (The MathWorks Inc. 2019, Natick, MA, USA) were utilized to calibrate the models and to test and validate their predictive capabilities.

### 2.7. Examination of the Model Performance

The predicted versus measured values were plotted, and the prediction quality was estimated by calculating the root mean square error of prediction (RMSEP) and relative root mean square error of prediction (pRMSEP), which were calculated using Equations (1) and (2), respectively. The coefficient of determination (R^2^) was calculated with Equation (3).
(1)$$\mathrm{RMSEP}=\sqrt{\frac{\sum_{i=}^{N}{\left(m_{i}-p_{i}\right)}^{2}}{N}}$$
(2)$$\mathrm{pRMSEP}\;\left[\%\right]=\frac{\mathrm{RMSEPx}100}{\max}$$

N, number of measurements; m_i_, measured values; p_i_, predicted values; max, maximum measured value; i, running index.
(3)$$R^{2}=1-\frac{\mathrm{RSS}}{\mathrm{TSS}}$$

RSS, residual sum of squares; TSS, total sum of squares.

### 2.8. Statistical Analysis

To build calibration and prediction models, Unscrambler X version 10.3 (CAMO Software AS., Oslo, Norway) and MATLAB R2019a version 9.6 (The MathWorks Inc. 2019, Natick, MA, USA) were utilized. Graphs were sketched with the same version of Matlab, which was used for model calibrations.

## 3. Results and Discussion

### 3.1. Off-Line Measurement of Cell Growth, Glucose and Lactic Acid

Off-line measured results of LPA6 and LCGG cell growth and concentrations of glucose and lactic acid are shown in Figure 1 and Figure 2, respectively. The fermentation of the teff-based substrate (hereinafter ‘substrate’) inoculated with mixed strains of LPA6 and LCGG predominantly had glucose (consumed substrate) and lactic acid (produced metabolite) [6]. Under condition 1, LPA6 and LCGG growth did not decline over fermentation time. Under condition 2, however, growth of both microbes began declining after 13 h of fermentation (Figure 1). Under condition 1, LPA6 and LCGG growth increased from 6 log cfu/mL to 8.49 and 8.29 log cfu/mL, respectively. Conversely, under condition 2, growth of LPA6 and LCGG decreased between 13 to 15 h fermentation from 8.24 to 8.21 log cfu/mL and from 7.99 to 7.74 log cfu/mL, respectively. At this decreasing point of cell growth, the main substrate glucose was not consumed entirely. Thus, the reason for declining cell growth appears to be due to the development of an acidic environment.

Under both fermentation conditions, LPA6 and LCGG cell growth was over 8 log cfu/mL. However, lower LCGG cell counts (7.74 log cfu/mL) were observed under condition 2. Both LPA6 and LCGG grew beyond the minimum level of the recommended viable probiotic of 6 log cfu/mL [18]. The minimum cell counts of probiotics must be achieved at the time of consumption to assure the probiotic effect of the product. The concentration of glucose and lactic acid changed from 1442.5 to 0 mg/L and 14.5 to 1831.5 mg/L, respectively. Overall, consumption of glucose by LPA6 and LCGG was associated with cell growth and lactic acid production (Figure 1 and Figure 2).

### 3.2. On-Line Measurement Using 2D-Fluorescence Spectroscopy

Under two different conditions after 0, 9, and 15 h fermentations, 2D-fluorescence spectra of substrate fermentation inoculated with mixed strains of LPA6 and LCGG are presented in Figure 3 and Figure 4. As can be seen, all fluorescence spectra showed peaks in the same region. High fluorescence intensities were observed in the region of excitation (410–450 nm) and emission (510–570 nm), where riboflavin typically reaches its maximum fluorescence [19,20,21]. The other peak region was visible at excitation (350–390 nm) and emission (430–490 nm), which showed the presence of NADH [22,23,24]. Moreover, all fluorescence spectra revealed peaks in the region of excitation (270–290 nm) and emission (310–390 nm), which verified the presence of protein [19,23,24,25].

For a better understanding of the fluorescence spectra variations, difference spectra were calculated by subtracting the initial spectrum from the spectra of 9 and 15 h fermentations (Figure 5 and Figure 6). The highest peak difference was observed in the protein fluorescence region at excitation (270–290 nm) and emission (310–390 nm) for all fluorescence spectra in 9 h fermentation. The other highest peak differences were observed in the fluorescence regions of riboflavin at excitation (410–450 nm) and emission (510–570 nm) and NADH for all fluorescence spectra in 15 h fermentation. The difference fluorescence spectra under condition 1 had a higher intensity as compared to the difference fluorescence spectra under condition 2. The difference and original fluorescence spectra showed peaks in the same regions. However, different fluorescence intensities were observed. These variations in the fluorescence intensities are due to differences in substrate concentration used under the two different fermentation conditions. The fluorescence intensities decreased from their initial values during the 9 h fermentation in all fluorescence regions. However, the fluorescence intensities increased during the 15 h fermentation in the regions of riboflavin and NADH.

The fluorescence intensities of NADH and riboflavin decreased during the exponential growth phase of LPA6 and LCGG. Conversely, their fluorescence intensities increased in the stationary phase of LPA6 and LCGG. This shows that riboflavin consumption is inversely associated with cell growth, meaning that it is consumed during the logarithmic phase of LPA6 and LCGG, but accumulated during a stationary phase. The accumulation of NADH begins when it no longer oxidizes to form the non-fluorophore molecule (NAD^+^), which results in an increase in its fluorescence intensity [3]. All fluorescence intensities decreased throughout the fermentation time in the protein region.

LCGG does not produce riboflavin [26]. Therefore, the production of riboflavin shown in the results can be attributed to LPA6 or the interaction effect of LPA6 and LCGG. Thakur and Tomar [27] reported riboflavin production ability of *Lactiplantibacillus plantarum* in MRS media. Riboflavin (vitamin B2) is a water-soluble vitamin and is essential for human health. It must be supplemented externally from food sources since it cannot be produced in the human body [28]. Hence, it is better to use bacteria, which can produce riboflavin rather than consume it throughout fermentation [26]. Thus, the capacity of mixed strains of LPA6 and LCGG to produce riboflavin can, together with their probiotic properties, be exploited for manufacturing multifunctional foods.

### 3.3. Prediction of Cell Growth in the Fermentation Process

The PLSR and ANN models were built using 1670 calibration samples. Off-line measured data and the corresponding 2D-fluorescence spectra were used to develop a model for predicting LPA6 and LCGG cell growth. Two different fermentation process conditions were examined for collecting on-line data as well as the corresponding off-line results. The prediction models for LPA6 and LCGG cell counts under condition 1 and condition 2 are presented in Figure 7, Figure 8, Figure 9 and Figure 10. The PLSR and ANN models were built separately for the prediction of LPA6 and LCGG cell counts.

Analysis of Cell Growth Using Artificial Neural Networks

Analysis of Cell Growth Using Partial Least-Squares Regression

Models and predictions for LPA6 and LCGG cell growth under two different fermentation conditions were performed using a maximum of six principal components. The RMSEP, pRMSEP, and R^2^ calculated for PLSR and ANN models are shown in Table 1. Better predictions were obtained with ANN for LCGG cell growth under condition 1. However, the predictions with PLSR were found to be better compared to ANN under condition 2. Predictions of principal component regression showed the highest rate of errors (data not shown) as compared to PLSR and ANN.

For the prediction of LPA6 and LCGG cell growth, the observed R^2^ values varied between 0.78–0.95, with the lowest R^2^ value being 0.78 for the prediction of LPA6 under condition 2 using ANN. In the prediction of the PLSR model, the pRMSEP was 3.7 and 3.9%, respectively, to predict LPA6 and LCGG cell growth under condition 1. Under the same condition, the ANN model had a pRMSEP of 2.5% to predict LPA6 cell growth and 2.5% to predict LCGG cell growth. Furthermore, the prediction of the PLSR model had a pRMSEP of 2.7 and 2.4% to predict LPA6 and LCGG cell growth, respectively, under condition 2. Under similar conditions, the ANN model had a pRMSEP of 4.5% to predict LPA6 cell growth and 3.6% to predict LCGG cell growth (Table 1). The lower pRMSEP and higher R^2^ values showed that the PLSR and ANN models were important for predicting LPA6 and LCGG cell counts in the fermentation of a teff-based substrate.

### 3.4. Prediction of Lactic Acid and Glucose in the Fermentation Process

The PLSR and ANN models were built using 1670 calibration samples. Lactic acid and glucose concentrations measured in an experiment were used with their corresponding on-line data to develop the PLSR and ANN models. To compare the PLSR and ANN models as well as to verify their performance, RMSEP, pRMSEP, and R^2^ values were calculated between the predicted and measured values (Table 2). The developed PLSR and ANN models were then used to predict lactic acid and glucose in another fermentation process run using 2D-fluorescence spectra. In principle, a direct measurement of glucose concentration by using fluorescence information is not possible, as it is not a fluorescence molecule. However, its consumption is directly related to an accumulation of fluorescence molecules such as tryptophan. Thus, it is possible to indirectly measure glucose by using 2D-fluorescence spectroscopy [29].

Analysis of Lactic Acid and Glucose Using Partial Least-Squares Regression

Prediction models for concentrations of glucose and lactic acid were developed using a maximum of seven principal components. The calculated results of RMSEP, pRMSEP, and R^2^ for the prediction using PLSR and ANN are shown in Table 2. Glucose and lactic acid prediction models are presented in Figure 11 and Figure 12. There were no significant differences observed between the prediction abilities of PLSR and ANN for glucose and lactic acid. Prediction of glucose and lactic acid using principal component regression showed the highest rate of errors (data not shown) as compared to PLSR and ANN. Overall, a stronger correlation was observed between 2D-fluorescence data and experimentally determined values of lactic acid than between 2D-fluorescence data and experimentally determined values of glucose.

Analysis of Glucose and Lactic Acid Using Artificial Neural Networks

Once the models were developed, it was possible to obtain results pertaining to cell counts, lactic acid, and glucose in minutes by using a two-dimensional fluorescence spectroscopy. However, it took more than three days to obtain the same results by using a plate count agar and high-performance liquid chromatography. For the analysis of lactic acid and glucose, we used expensive instruments such as a deep freezer, centrifugation, pump, filter, and fully equipped high-performance liquid chromatography. This form of analysis is time-consuming and labor-intensive. To determine LPA6 and LCGG cell counts using a plate count agar, we used several chemicals and spent a long time performing tedious work. However, without using the instruments required for the conventional analysis, similar results were obtained in minutes by using a two-dimensional fluorescence spectroscopy integrated with PLSR and ANN.

## 4. Conclusions

An essential vitamin (riboflavin) was accumulated in the fermentation of a teff-based substrate inoculated with mixed strains of LPA6 and LCGG. The riboflavin production ability of LPA6 and LCGG, together with their probiotic properties, could be exploited for manufacturing multifunctional food products. A 2D-fluorescence spectroscope is an ideal instrument for the rapid supervision of the fermentation process without interfering with the fermentation medium. It provides broad information about metabolic changes occurring during the fermentation process. This study has shown that 2D-fluorescence spectroscopy coupled with PLSR and ANN models can be applied to accurately monitor LPA6 and LCGG cell counts and lactic acid concentration in the fermentation of a teff-based substrate. It might even be possible to use a simple, inexpensive fluorescence sensor using light-emitting diodes and photodiodes.

## Figures and Tables

**Figure 1 foods-11-01171-f001:**
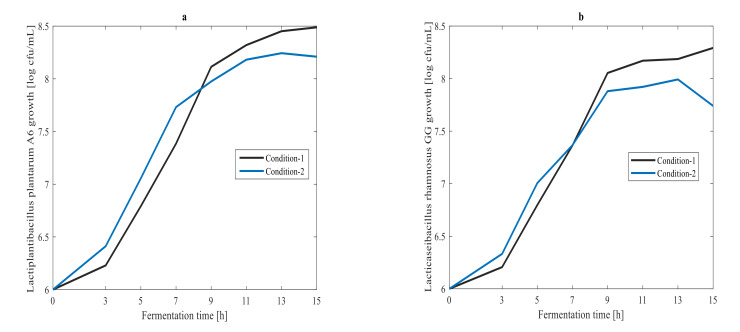
(**a**) *Lactiplantibacillus plantarum* A6 and (**b**) *Lacticaseibacillus rhamnosus GG* growth under two different fermentation conditions: Condition 1, 7 g/100 mL substrate inoculated with 6 log cfu/mL mixed strains of *Lactiplantibacillus plantarum* A6 and *Lacticaseibacillus rhamnosus* GG; Condition 2, 4 g/100 mL substrate inoculated with 6 log cfu/mL mixed strains of *Lactiplantibacillus plantarum* A6 and *Lacticaseibacillus rhamnosus* GG.

**Figure 2 foods-11-01171-f002:**
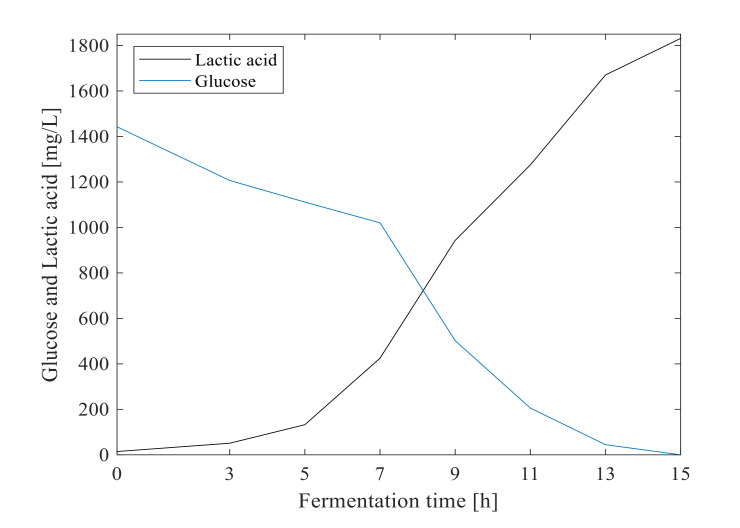
Glucose consumption and Lactic acid production during fermentation of 7 g/100 mL substrate inoculated with 6 log cfu/mL mixed strains of *Lactiplantibacillus plantarum* A6 and *Lacticaseibacillus rhamnosus* GG.

**Figure 3 foods-11-01171-f003:**
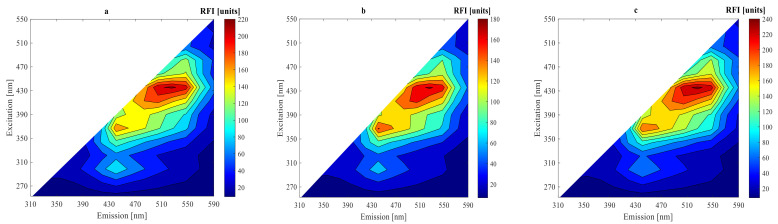
Original spectra after (**a**) 0 h, (**b**) 9 h, and (**c**) 15 h fermentation of 7 g/100 mL substrate inoculated with 6 log cfu/mL mixed strains of *Lactiplantibacillus plantarum* A6 and *Lacticaseibacillus rhamnosus* GG.

**Figure 4 foods-11-01171-f004:**
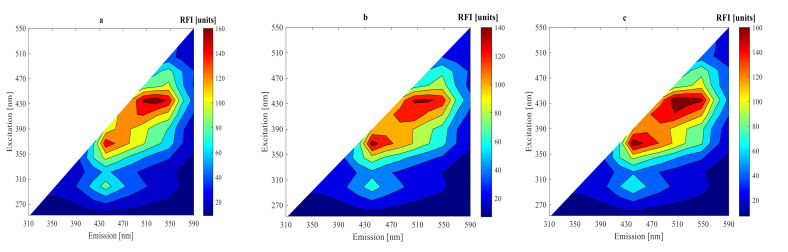
Original spectra after (**a**) 0 h, (**b**) 9 h, and (**c**) 15 h fermentation of 4 g/100 mL substrate inoculated with 6 log cfu/mL mixed strains of *Lactiplantibacillus plantarum* A6 and *Lacticaseibacillus rhamnosus* GG.

**Figure 5 foods-11-01171-f005:**
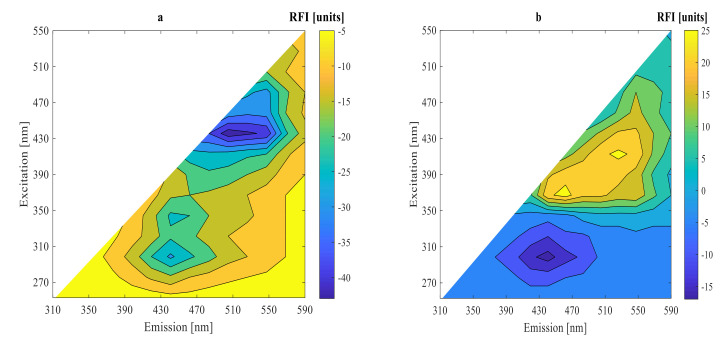
Subtracted spectra of the initial one from the original spectra after (**a**) 9 h and (**b**) 15 h fermentation of 7 g/100 mL substrate inoculated with 6 log cfu/mL mixed strains of *Lactiplantibacillusplantarum* A6 and *Lacticaseibacillusrhamnosus* GG.

**Figure 6 foods-11-01171-f006:**
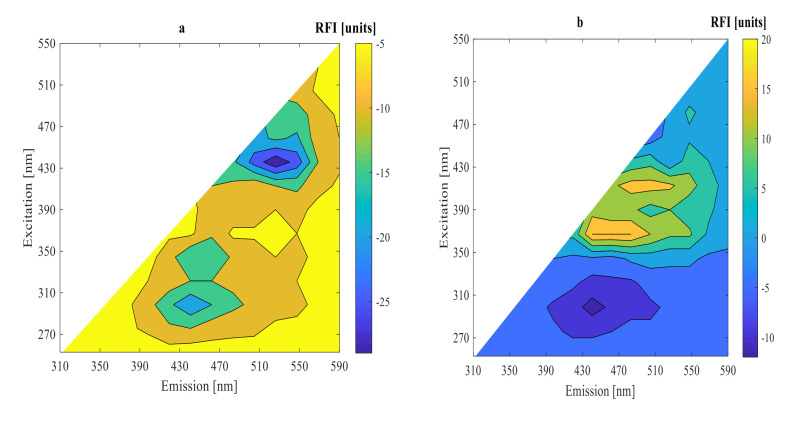
Subtracted spectra of the initial one from the original spectra after (**a**) 9 h and (**b**) 15 h fermentation of 4 g/100 mL substrate inoculated with mixed strains of *Lactiplantibacillusplantarum* A6 and *Lacticaseibacillusrhamnosus* GG.

**Figure 7 foods-11-01171-f007:**
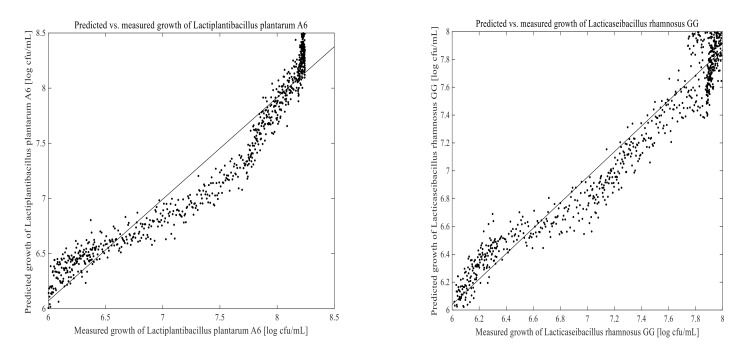
Predicted vs. measured cell growth in the fermentation of 4 g/100 mL of substrate inoculated with mixed strains of 6 log cfu/mL *Lactiplantibacillus plantarum* A6 and *Lacticaseibacillus rhamnosus* GG; predicted with partial least-squares regression using four principal components.

**Figure 8 foods-11-01171-f008:**
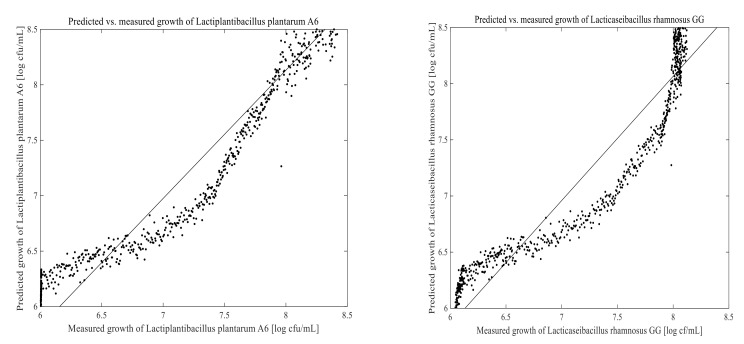
Predicted vs. measured cell growth in the fermentation of 7 g/100 mL of substrate inoculated with mixed strains of 6 log cfu/mL *Lactiplantibacillus plantarum* A6 and *Lacticaseibacillus rhamnosus* GG; predicted with partial least-squares regression using six principal components.

**Figure 9 foods-11-01171-f009:**
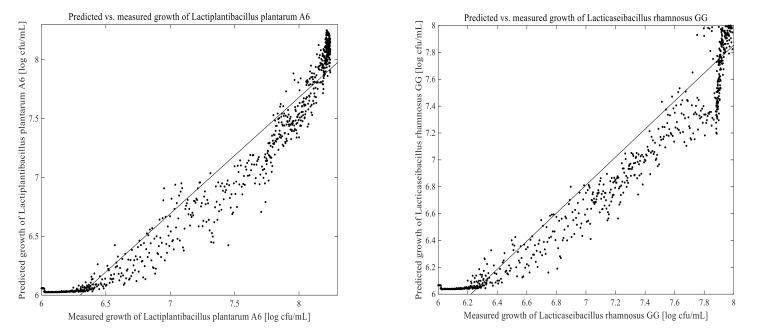
Predicted vs. measured cell growth in the fermentation of 4 g/100 mL of substrate inoculated with mixed strains of 6 log cfu/mL *Lactiplantibacillus plantarum* A6 and *Lacticaseibacillus rhamnosus* GG; predicted with artificial neural networks using five hidden neurons.

**Figure 10 foods-11-01171-f010:**
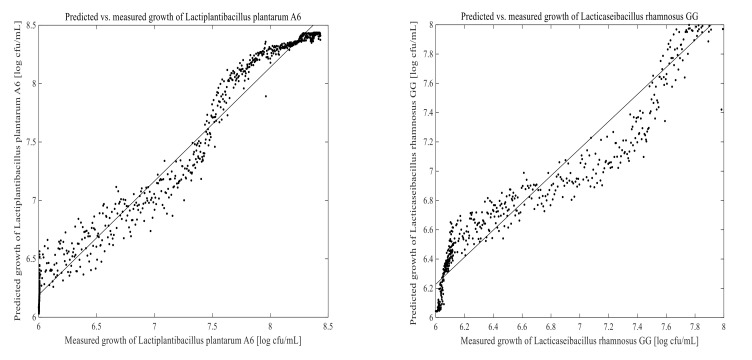
Predicted vs. measured cell growth in the fermentation of 7 g/100 mL of substrate inoculated with mixed strains of 6 log cfu/mL *Lactiplantibacillus plantarum* A6 and *Lacticaseibacillus rhamnosus* GG; predicted with artificial neural networks using five hidden neurons.

**Figure 11 foods-11-01171-f011:**
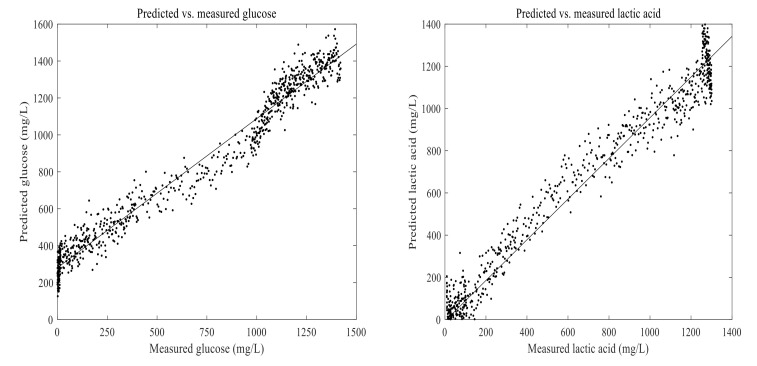
Predicted vs. measured glucose and lactic acid in the fermentation of 7 g/100 mL of substrate inoculated with 6 log cfu/mL mixed strains of *Lactiplantibacillus plantarum* A6 and *Lacticaseibacillus rhamnosus* GG; predicted with partial least-squares regression using seven principal components.

**Figure 12 foods-11-01171-f012:**
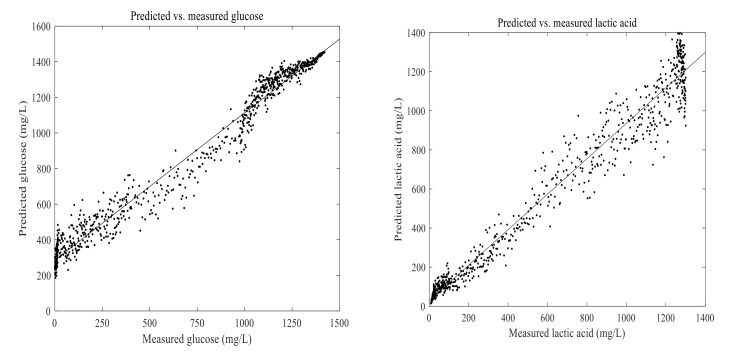
Predicted vs. measured glucose and lactic acid in the fermentation of 7 g/100 mL of substrate inoculated with 6 log cfu/mL mixed strains of *Lactiplantibacillus plantarum* A6 and *Lacticaseibacillus rhamnosus* GG; predicted with artificial neural networks using one hidden neuron.

**Table 1 foods-11-01171-t001:** The RMSEP, pRMSEP, and R^2^ values for the prediction of LPA6 and LCGG cell growth using PLSR and ANN under two different fermentation conditions.

Predicted with PLSR	Condition 1			Condition 2		
	RMSEP (log cfu/mL)	pRMSEP (%)	R^2^	RMSEP (log cfu/mL)	pRMSEP (%)	R^2^
LPA6	0.31	3.7	0.88	0.22	2.7	0.92
LCGG	0.32	3.9	0.85	0.20	2.4	0.92
Predicted with ANN	RMSEP (log cfu/mL)	pRMSEP (%)	R^2^	RMSEP (log cfu/mL)	pRMSEP (%)	R^2^
LPA6	0.21	2.5	0.95	0.37	4.5	0.78
LCGG	0.20	2.5	0.94	0.29	3.6	0.83

LPA6, *Lactiplantibacillus plantarum* A6; LCGG, *Lacticaseibacillus rhamnosus* GG; RMSEP, root mean square error of prediction; pRMSEP, relative root mean square error of prediction; R^2^, coefficient of determination; PLSR, partial least-squares regression; ANN, artificial neural network; Condition 1, 7 g/100 mL substrate inoculated with 6 log cfu/mL; Condition 2, 4 g/100 mL substrate inoculated with 6 log cfu/mL.

**Table 2 foods-11-01171-t002:** RMSEP, pRMSEP, and R^2^ values for the prediction of glucose and lactic acid with PLSR and ANN during the fermentation process of 7 g/100 mL of substrate inoculated with 6 log cfu/mL mixed strains of LPA6 and LCGG.

Analyte	PLSR			ANN		
	RMSEP (log cfu/mL)	pRMSEP (%)	R^2^	RMSEP (log cfu/mL)	pRMSEP (%)	R^2^
Glucose	191.72	13.5	0.86	199.92	14.1	0.85
Lactic acid	98.41	7.6	0.96	100.09	7.7	0.96

LPA6, *Lactiplantibacillus plantarum* A6; LCGG, *Lacticaseibacillus rhamnosus* GG; RMSEP, root mean square error of prediction; pRMSEP, relative root mean square error of prediction, R^2^, coefficient of determination; PLSR, partial least-squares regression; ANN, artificial neural network.

## Data Availability

Data is contained within the article.

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
