# Peer review of "Application of Two-Dimensional Fluorescence Spectroscopy for the On-Line Monitoring of Teff-Based Substrate Fermentation Inoculated with Certain Probiotic Bacteria"

_foods, 2022, doi:10.3390/foods11081171_

Round 1
Reviewer 1 Report
This manuscript is extremely interesting and timely. It is thematically relevant to the thematic scope of the journal, so I believe that it should be allowed to be published. However, before that it has to be corrected and supplemented with missing information:
- Taxonomy of lactobacilli – please pay attention to the current systematics of lactic acid bacteria (lactobacilli) and use their current nomenclature in the manuscript (see: http://lactobacillus.ualberta.ca/)
- Section „ Materials and Methods” – please provide information on the number of repetitions of individual experiments and the number of repetitions of individual analyzes.
- Line 116 – please provide information on how long inoculum were held at 6 °C.
- Section „2.4. Fermentation Process Condition” – please justify the choice of the fermentation parameters used. Why 4 and 7 g/100 139 mL of whole grain teff flour in distilled water? Why different portions of inoculum? Does it have a technological justification?
Author Response
Dear Reviewer,
We have revised the manuscript intensively as per your comments. Some points are mentioned just after your statements with track changes function. A single document is attached here.
Thank you for your important comments.

Reviewer 2 Report
The authors presented the fermentation of teff based flour with two probiotic strains as in a previous paper by the same authors.
The research is aimed to offer a way to analyse and monitor the fermentation process that is not time consuming, costly, and labor intensive. For this reason they apply two-dimensional (2D) fluorescence spectroscopy coupled with partial least squares regression (PLSR) and artificial neural network (ANN) for the on-line quantitative monitoring of cell growth, lactic acid, and glucose concentra tion in the fermentation of a teff-based substrate. However no information is given on the time, cost or labour needed to get the results through this technique, nor comparison with usually applied methods.
The inoculum concentration is said to be of 6 log/ml, but the dilution from the original 9 log/ml and the respective amount of the two strains is missing.
The method used to selectively count LA6 and LGG is not clear.
The substrate is heated to 85°C and then sterilized. How is the sterilization achieved and controlled?
Fermentation process conditions are not well described (it can be desumed from the result section but it shoul be described better in the M&M section).
The count of strains is reported to be beyond the minimum level of the recommended viable probiotic of 6 log cfu/mL, however, how is the fermented flour going to be consumed? raw?
Considering that teff is made mainly by starch, the amount of glucose is maybe not so informative on the fermentation trend. At least, a composition of the substrate should be given before the fermentation.
The riboflavine is said to be consumed in the first steps and remaining after 13 hours, but afterward authors conclude that the fermentation allow for the accumulation of riboflavine.
The link with treatment of S. aureus does not fit the discussion.
Are the fluorescent intensities univoque for the compounds the authors consider (riboflavine, NADH and protein)?
Lactobacillus rhamnosus is now Lacticaseibacillus rhamnosus, Lactobacillus plantarum is now Lactiplantibacillus plantarum.
Author Response

(The authors gave the same response as above.)

Reviewer 3 Report
The work described here is related to Application of Two-Dimensional Fluorescence Spectroscopy for the On-Line Monitoring of Teff-Based Substrate Fermentation Inoculated with Certain Probiotic Bacteria. Good Work, May be accepted with minor changes.
Abstract: The useful information should be presented, therefore, the abstract needs to rewrite.
The framing of sentences should be simple, crisp and to the point.
English needs a little improvement
Data analysis methods should be added
Author Response

(The authors gave the same response as above.)

Round 2
Reviewer 2 Report
I appreciate the revision made by the authors but I still dound that some points has to be clarified.
Authors refer to a method used to selectively count LA6 and LGG already published. However, no information were reported in the previous work on how the effective selective evaluation of the two strains was checked. Colonies have different morphology? Do they check the colonies by microscopic visualization? How do they can confirm that the counts are differentiated just based on different hours of incubation?
Authors replied to my former observation: "However no information is given on the time, cost or labour needed to get the results through this technique, nor comparison with usually applied methods". Some of the information given in the reply should be included in the text.
Similarly, they replied to the observation: The count of strains is reported to be beyond the minimum level of the recommended viable probiotic of 6 log cfu/mL, however, how is the fermented flour going to be consumed? raw?
I was meaning that if the fermented teff is going to be consumed as cooked, the viability of cells will anyway be lost. How do you comment on this?
The amount of glucose before fermentation should be given
authors replied that "at the end of the fermentation time the riboflavin content was increased as compared to its concentration at the beginning of the fermentation. However, its concentration was reduced in the intermediate" How they comment on this?
If the fluorescent intensities are not univoque for the determed molecules, how can the authors conclude that the increase/decrease of fluorescence is due to consumption/production of those molecules?
There are several typos in the text, that has to be carefully read and corrected.
Author Response
Dear Reviewer,
The English language and style of the manuscript has been revised by a native English professional. All your expert comments and questions are considered point by point. A document with responses to your comments is uploaded here.
Thank you for your comments.
